# Phytopathogenic Fungicidal Activity and Mechanism Approach of Three Kinds of Triphenylphosphonium Salts

**DOI:** 10.3390/jof10070450

**Published:** 2024-06-27

**Authors:** Xuelian Liu, Huihui Liu, Fahong Yin, Yiyi Li, Jiazhen Jiang, Yumei Xiao, Yanhua Wu, Zhaohai Qin

**Affiliations:** 1College of Science, China Agricultural University, Beijing 100193, China; lxl_nuo@126.com (X.L.); liuhuihui@meibang.cn (H.L.); yinfahong@cau.edu.cn (F.Y.); yiyi.li@cau.edu.cn (Y.L.); jiangjiazhen@cau.edu.cn (J.J.); xiaoyumei@cau.edu.cn (Y.X.); wuyanhua@cau.edu.cn (Y.W.); 2School of Pharmaceutical Sciences and Institute of Materia Medica, Xinjiang University, Urumqi 830017, China

**Keywords:** triphenylphosphonium salt (TPP), fungicidal activity, cut-off effect, phytotoxicity

## Abstract

The triphenylphosphonium (TPP) cation has been widely used as a carrier for mitochondria-targeting molecules. We synthesized two commonly employed targeting systems, namely, ω-triphenylphosphonium fatty acids (group 2) and ω-triphenylphosphonium fatty alcohols (group 3), to assess the impact of the TPP module on the biological efficacy of mitochondria-targeting molecules. We evaluated their fungicidal activities against nine plant pathogenic fungi in comparison to alkyl-1-triphenylphosphonium compounds (group 1). All three compound groups exhibited fungicidal activity and displayed a distinct “cut-off effect”, which depended on the length of the carbon chain. Specifically, group **1** compounds showed a cut-off point at C_10_ (compound **1**–**7**), while group **2** and **3** compounds exhibited cut-off points at C_15_ (compound **2**–**12**) and C_14_ (compound **3**–**11**), respectively. Notably, group **1** compounds displayed significantly higher fungicidal activity compared to groups **2** and 3. However, group **2** and **3** compounds showed similar activity to each other, although susceptibility may depend on the pathogen tested. Initial investigations into the mechanism of action of the most active compounds suggested that their fungicidal performance may be primarily attributed to their ability to damage the membrane, as well as uncoupling activity and inhibition of fungal respiration. Our findings suggest that the TPP module used in delivery systems as aliphatic acyl or alkoxyl derivatives with carbon chains length < 10 will contribute negligible fungicidal activity to the TPP-conjugate compared to the effect of high level of accumulation in mitochondria due to its mitochondria-targeting ability. These results provide a foundation for utilizing TPP as a promising carrier in the design and development of more effective mitochondria-targeting drugs or pesticides.

## 1. Introduction

Triphenylphosphonium (TPP) is commonly employed in the Wittig reaction and has recently attracted significant attention for its applications in materials science [1] and medicinal chemistry [2]. Quaternary phosphonium salts (QPS), including TPP, are characterized by their high liposolubility (hydrophobicity) and positive charge, facilitating their ability to penetrate the phospholipid bilayer without the need for transporters [3,4]. Consequently, TPP is often favored as a carrier for drugs that target mitochondria and bacteria [5], although some studies have raised doubts about this perspective [6,7]. Combining TPP cations with established drugs such as doxorubicin, cisplatin, chlorambucil, and camptothecin typically enhances their cytotoxicity [2]. Previous research has demonstrated the superior antifungal properties of TPP-conjugated respiratory chain inhibitors compared to their original forms [8,9,10,11,12]. However, while the targeting and enhancing capabilities of TPP are known, the independent biological activity of the TPP module and its role in the overall bioactivity of conjugated molecules remain uncertain and subject to debate.

While the TPP module has traditionally been viewed as an inert mitochondria-targeting carrier [13], recent evidence suggests that it possesses inherent biological activities. For example, polyphenylene-ether-modified TPP has demonstrated antibacterial properties against *Staphylococcus epidermidis* and *Escherichia coli* [14]. Moreover, Taladriz et al. reported the antitrypsin activity of benzyl-TPP against Trypanosoma brucei [15], and Bergeron discovered that TPP compounds can inhibit DNA amplification in vitro at concentrations below the aggregation threshold. The interaction between TPP and DNA is strongly linked to bacterial toxicity, indicating that TPP molecules may disrupt DNA metabolism in cells [16]. Additionally, in a study where mice were administered C_12_TPP at a dose of 50 μmol/kg b.w. per day in drinking water, a 12% reduction in body weight and a 24% decrease in body fat mass were observed within the first seven days of treatment [17]. Notably, this treatment did not affect water taste or consumption and had no impact on the fat and energy content in feces [17]. QPS have emerged as a promising class of versatile organic preservatives with several advantages, including low foaming properties, high sludge-removal capability, and broad pH range applicability. QPS have been extensively studied as active ingredients for the development of antibacterial materials [18,19].

In order to comprehensively explore the role of TPP in fungicidal activity in agriculture, we synthesized two commonly utilized targeting systems: ω-triphenylphosphonium fatty acids and ω-triphenylphosphonium fatty alcohols, as they can be easily conjugated with cargos through esterification. We assessed their bioactivities and mechanisms of action in vitro, comparing them with alkyl-1-triphenylphosphonium. Moreover, we examined the impact of the electronic properties of phenyl substituents on their activities (Figure 1). The findings of this investigation will establish a theoretical framework for the development of TPP-driven mitochondrial-targeting fungicides.

## 2. Experimental

### 2.1. Equipment and Materials

The ^1^H NMR and ^13^C NMR spectra were obtained using a Bruker AVANCE NEO 400 or 500 MHz spectrometer, with tetramethylsilane serving as the internal standard. Melting points were determined using a WRS-1C melting point apparatus and were not corrected. High-resolution mass spectra (HRMS) were obtained using an ESI-Q-TOF instrument. Chemicals and reagents used were obtained from commercial sources and were not further purified.

### 2.2. Synthetic Procedures

General Synthetic Procedures of Compounds **1**. A solution of 15 mmol triphenylphosphine and 10 mmol 1-bromoalkane in 50 mL acetonitrile was refluxed for 24 h. Afterward, the mixture was evaporated to dryness and the resulting residue was subjected to purification by silica gel column chromatography using a mixture of petroleum ether, ethyl acetate, and methanol in a ratio of 15:15:1 (*V*/*V*/*V*). This purification process afforded compounds **1**.

General Synthetic Procedures of Compounds **1A**–**1C**. A 50 mL solution of acetonitrile, containing 15 mmol of tri-(4-substitutedphenyl)phosphine and 10 mmol of 1-bromoalkane, was refluxed for 24–72 h. The resulting mixture was then evaporated to dryness. The residue was purified using silica gel column chromatography, with an eluent mixture of petroleum ether, ethyl acetate, and methanol in a ratio of 15:15:1 (-*V*/*V*/*V*). This purification method yielded three types of compounds, namely, **1A**, **1B**, and **1C**. 

General Synthetic Procedures of Compounds **2**. A 50 mL acetonitrile solution containing 15 mmol tri-(4-substitutedphenyl)phosphine and 10 mmol ω-bromoalkanoic acid was refluxed for 24 h and then evaporated to dryness. The residue was purified by silica gel column chromatography (petroleum ether: ethyl acetate: methanol = 15:15:1) to afford compounds **2**.

Synthesis of **2**–**12′**. A total of 3.11 mmol of 15-bromopentadecanoic acid was dissolved in 45 mL of DCM, and 3.73 mmol of oxalyl chloride and two drops of DMF were added. The mixture was stirred on an ice bath for 0.5 h. After evaporating the solvent to dryness, 1.06 g of 15-bromopentadecanoyl chloride was obtained.

A 45 mL DCM solution containing 4.68 mmol of methanol was added to the residue, and the mixture was stirred for 30 min at room temperature. The reaction mixture was then washed three times with water and concentrated under vacuum. The resulting residue was redissolved in acetonitrile and 3.20 mmol of triphenylphosphine was added. The mixture was refluxed for 24 h with continuous stirring. After evaporating the solvent under vacuum, the residue was purified by silica gel column chromatography using an eluent with a mixture of ethyl acetate and methanol in a 10:1 (*V*/*V*) ratio. Compound **2**–**12′** was obtained as a viscous oil with a 60% yield. 

General Synthetic Procedures of Compounds **3**. A 50 mL solution of acetonitrile containing 15 mmol of triphenylphosphine and 10 mmol of ω-bromoalkan-1-ol was subjected to reflux for 24 h and then evaporated to dryness. The resulting residue was subsequently purified using silica gel column chromatography with a mixture of petroleum ether, ethyl acetate, and methanol in a ratio of 15:15:1 (*V*/*V*/*V*). This procedure yielded compounds **3**. 

Synthesis of **3**–**11′**. A 45 mL solution of DCM containing 3.77 mmol of 12--romo-dodecan-1-ol was mixed with 4.52 mmol of acetyl chloride while stirring. After 30 min, the solvent was removed under reduced pressure, and the remaining substance was dissolved in 30 mL of acetonitrile. To this solution, 4.54 mmol of triphenylphosphine was added. The mixture was refluxed for twenty-four hours and then evaporated using vacuum. The resulting residue was purified using silica gel column chromatography with a mixture of ethyl acetate and methanol in a ratio of 10:1. The purified compound, named as **3**–**11′**, was obtained as a colorless oil with a yield of 64%. The ^1^H NMR, ^13^C NMR, and HRMS spectrum data of all synthesised compounds were shown in File S1.

In Vitro Fungicidal Activity Assay [20]. The in vitro fungicidal activity of the target compounds was evaluated against nine plant pathogens using the mycelia growth inhibition method. The target compound was dissolved in DMSO to prepare a stock solution, which was then mixed with PDA medium to prepare a medicated medium with a DMSO content of 1‰ and a title compound concentration of 70 µM. And PDA medium with 1‰ DMSO was used as control. After the culture solidified, punch holes (5 mm in diameter) were made at the outer edge of the active colony and inoculated in the center of the medicated culture medium. The Petri dishes were then placed in an incubator at 25 °C for inverted cultivation. Once the colony diameter of the control group reached four-fifths of that of the Petri dish, the colony diameter was measured using a cross method and recorded. Each experiment was performed in three times replicates. The relative inhibition ratio (%) was calculated using Equation (1):The relative inhibition ratio (%) = (colony diameter of control − colony diameter of treatment)/(colony diameter of control − 0.5cm) × 100%(1)

Spore Germination Inhibition Assay. Stock solutions were prepared by dissolving the target compounds or fluazinam in DMSO, then mixing them with water agar medium (WA medium) to create medicated plates with various concentrations, each containing 2‰ DMSO. The WA medium with 2‰ DMSO served as the control group, comprising glucose (20 g), agar (20 g), and deionized water (800 mL). A spore suspension of *P. capsici* at a concentration of 1 × 10^5^ pcs/mL (100 µL) was evenly distributed on the medicated plate and incubated in darkness at 25 °C. Each treatment was replicated three times. The spores in the control group were observed under an electron microscope until all spores had germinated, indicated by the bud tube length reaching the spore radius. The total number of spores and germinated spores were enumerated, and the inhibition rate of each drug treatment on spore germination was calculated using Equations (2) and (3). Data analysis was performed using IBM SPSS Statistics 26 software.
Spore germination rate (%) = (germinated number of spores/total number of spores) × 100%(2)
Inhibition rate (%) = [(spore germination rate of blank group − spore germination rate of treating group)/spore germination rate of blank group] × 100%(3)

Respiratory Inhibiting Assay [21]. Seven 0.5 cm *P. capsici* mycelia-agar plugs were introduced into 100 mL of sterilized potato dextrose broth (PDB) medium. The culture was incubated on a shaker at 25 °C and 175 rpm in the absence of light. After a 2 day incubation period, the plugs were extracted, and the mycelial surface was rinsed with deionized water. Subsequently, the mycelia underwent suction filtration, were weighed, and then were assessed using a Clark oxygen electrode.

In the reaction tank, 2 mL of a 1% glucose solution and different concentrations of tested compounds were added. The blank group received the same amount of DMSO. Once the baseline was stable (approximately 2 min), 40 mg fresh mycelia was added, and the oxygen consumption was observed and recorded.

Membrane Permeability: The Conductivity Method [22]. A total of 200 mg of *P. capsici* hyphae, which were cultured in a potato dextrose broth (PDB) and filtered using a vacuum pump, were suspended in 20 mL of sterile water containing a 1% glucose solution. Next, a tested compound was added to achieve a final concentration of 17.5 μM (containing 2‰ DMSO). The samples were then cultured at 25 °C in a constant temperature shaker at 175 rpm. The conductivities of the mixtures were measured using a conductivity meter (DDS-11A) at 0, 1, 3, 5, 7, and 9 h, individually. After 9 h, all samples were subjected to a 30 min boiling treatment to induce cell death, followed by cooling to room temperature and measurement of the final electric conductivities. The relative leakage of hyphae was quantified according to Equation (4): Relative leakage (%) = [(conductivity at a certain moment − initial electric conductivity)/final conductivity] × 100%(4)

Scanning Electron Microscopy (SEM) Assay [23]. A fungus plug measuring 5 mm in diameter was taken from the edge of the *P. capsici* fungus colony on PDA medium containing 35 μM of the tested compounds. The sample was then fixed in 2.5% glutaraldehyde at 4 °C for 24 h, washed with 0.1 M phosphate buffer for 15 min, and subsequently fixed in a 1% OsO_4_ solution for an additional hour. The samples were dehydrated using a gradient of ethanol (20%, 50%, 80%, and 100%). Finally, the samples were air-dried, mounted on aluminum stubs, coated with gold using a sputter coater, and examined using a scanning electron microscope (SU8100). 

Crop Safety Assay. Pepper leaves displaying consistent growth were collected and subjected to two rounds of spraying with an aqueous solution at a compound concentration of 100 μg/mL and a DMSO content of 1‰. Subsequently, the leaves were cultured for 7 days in a sterile Petri dish containing moist filter paper at the bottom. The phytotoxic symptoms, including leaf withering, lesions, and perforation, were visually observed. A water solution with 1‰ DMSO content was used as the blank control, while a 100 μg/mL fluazinam aqueous solution was used as the positive control. Similarly, rape leaves were treated in the same manner, but with either a 100 or 200 μg/mL solution, and cultured for 24 h. 

## 3. Results and Discussion

### 3.1. Chemistry

With the exception of esters **2**–**12′** and **3**–**11′**, all compounds were readily synthesized via nucleophilic substitution of triphenylphosphine with brominated starting materials. The electronic impact of substituents on the benzene ring significantly influenced the yield. For example, tris(4-trifluoromethyl-phenyl)phosphine only produced a 10% yield of the corresponding phosphonium compound, even when the reaction time was extended to 24 h. Nevertheless, we did not optimize the reaction conditions as it was deemed non-essential.

### 3.2. In Vitro Fungicidal Activity

The synthesized compounds were first screened for activity against nine plant pathogenic fungi at a concentration of 70 μM. The results are outlined in Table 1, demonstrating significant variability in sensitivity to these compounds among the various fungi. Moreover, the chain length of compounds within the same series was found to have a significant impact on their activity.

To analyze the relationship between the structure and activity of these compounds, we classified and compared three different compound types (Figure 2). The activity against all pathogenic fungi illustrated a consistent pattern of initially increasing from low to high and then decreasing with an increase in chain length (Figure 2A–C), reminiscent of the well-documented “cut-off effect” observed in long-chain surfactants over 80 years ago [24,25]. These compounds share similarities with quaternary ammonium salts, and their fungicidal activity is likely attributed to interactions with the cell membrane [26]. Importantly, the sensitivity of pathogenic fungi to these three compound types varies significantly. Notably, at a concentration of 70 μM, type **1** compounds with the same carbon chain length displayed remarkable fungicidal activity against most of the tested pathogens, surpassing their corresponding type **2** and type **3** compounds (Figure 2E). The average inhibition rate against the nine pathogenic fungi served as the comprehensive evaluation index for fungicidal activity (Figure 2D), effectively illustrating the relationship between carbon chain length and fungicidal activity. For type **1** compounds, fungicidal activity increased rapidly with chain length until peaking at C_10_ (compound **1**–**7**), after which it sharply declined. Conversely, type **2** and **3** compounds displayed a different trend, with low activity observed before C10, followed by an increase beyond C_10_. Type **2** reached their cut-off point at C_15_ (compound **2**–**12**), while type **3** reached it at C_14_ (compound **3**–**12**), yet the activity peaks of the two compounds were nearly identical. These findings indicate that the presence of a hydroxyl or carboxyl group notably decreased the fungicidal activity of TPP salt, potentially reducing damage to the cell membrane. It is worth noting that the cut-off points may exhibit slight variations across different pathogens.

In order to evaluate the influence of carboxyl or hydroxyl groups on activity, esterification was conducted on **2**–**12** and **3**–**11**, generating **2**–**12′** and **3**–**11′**, respectively. Interestingly, the average inhibition rate of the esters decreased in comparison to **2**–**12** and **3**–**11,** which contained free hydroxyl groups, with **2**–**12′** declining by approximately 15.8% and **3**–**11′** decreasing by about 3.7%, despite a slight increase in activity against individual pathogens like RS and BC. This observation suggests that the contribution of TPP-acyloxyl and TPP-alkoxyl as mitochondrial target delivery systems to fungicidal activity is minimal when the chain length is less than C_10_. These outcomes lay a strong groundwork for the future development of mitochondria-targeting drugs using TPP as the target delivery system.

To investigate the impact of electronic effects of substituents on the benzene ring on activity, six compounds (**1A**–**1C**) were synthesized. When compared with compounds **1**–**7** and **1**–**8**, introducing an electron donor at the 4-position only slightly improved activity (Table 1). Notably, the methoxy group, with stronger electron-donating ability, exhibited slightly lower activity than the methyl group. Conversely, incorporating an electron-withdrawing trifluoromethyl group (**1C-1** and **1C-2**) drastically reduced activity, underscoring the crucial role of delocalization of positive charge of the phosphonium salt in determining its activity.

*P. capsici* is a significant pathogen known to cause phytophthora blight in pepper plants, and the severity of this disease has been increasing in recent years [27]. To determine the specific toxicity levels of compounds **1**–**6**, **1**–**7**, **1**–**8**, **1**–**9**, **2**–**12**, and **3**–**11** against *P. capsici*, additional assessments were carried out using the mycelial growth inhibition method, with the results presented in Table 2. Remarkably, compound **1**–**8** demonstrated exceptional efficacy comparable to that of fluazinam, suggesting its potential utility in the management of *P. capsici*.

### 3.3. Spore Germination Inhibition Activity

Since the germination of *P. capsici* conidia is dependent on energy supply and closely linked to energy metabolism, we conducted a study to assess the spore germination inhibition activity of three compounds against *P. capsici*. The results, presented in Table 3 and Figure 3, reveal that compounds **1**–**8**, **2**–**12**, and **3**–**11** showed modest inhibitory effects on the spore germination of *P. capsici*. These effects were notably weaker compared to fluazinam, suggesting that the fungicidal activity of these compounds may have little association with energy supply. This finding contrasts with our prior research, which demonstrated significant spore germination inhibition by TPP-conjugated respiratory chain complex inhibitors [8,9,10]. It implies that the TPP module likely plays a minor role in inhibiting energy synthesis in these compounds.

### 3.4. Influence on the Respiratory of P. capsici Mycelia

Damage to the mitochondrial membrane can lead to the release of protons, resulting in a decrease in the proton gradient across the membrane, directly impacting cellular respiration. Therefore, we explored the effect of compounds **1**–**8**, **2**–**12**, and **3**–**11** on the mycelial respiration of *P. capsici*, as depicted in Figure 4. Generally, these compounds exhibit characteristics akin to fluazinam, stimulating respiration at low concentrations and hindering it at high concentrations, resembling mitochondrial uncouplers. Interestingly, the promotion of mycelial respiration at low concentrations did not show dependency on concentration or time. However, compounds **1**–**8** and **3**–**11** displayed weak inhibitory activities at concentrations of 1.093 and 2.18 μM, respectively. Conversely, at elevated concentrations, all compounds showed a concentration-dependent and time-dependent relationship with mycelial respiration inhibition. Particularly noteworthy is that at a concentration of 35 μM, compound **1**–**8** achieved an inhibitory rate on mycelial respiration of 83.22%, surpassing that of fluazinam at 49.79%. At 70 μM, compound **2**–**12** inhibited mycelial respiration by 54.53%, slightly higher than fluazinam at the same concentration, whereas compound **3**–**11** exhibited a 28.80% inhibition rate at this concentration.

The uncoupling activity of Cn-TPP^+^ has been previously documented, with a proposed mechanism suggesting that Cn-TPP^+^ can interact with fatty acid anions to form an ion pair. This interaction facilitates the proton transport cycle of naturally weakly acidic uncoupling fatty acids, leading to an uncoupling effect and enabling proton transport across the mitochondrial membrane [28,29]. On the other hand, C_12_TPP^+^ demonstrates substantial inhibition of complexes I, III, and IV at 0.5 mM, with near-complete inhibition, while complex II is inhibited by around 50% at this concentration. Incorporation of alkyl TPP^+^ molecules into the mitochondrial inner membrane is believed to elevate proton leakage and inhibit the respiratory chain complex, disrupting its normal function [13]. Since the respiratory chain complexes rely on their lipid environment and necessitate phospholipid molecules for their operation [30,31], an excess of alkyl-TPP^+^ molecules in the membrane can compromise the membrane’s insulating properties, leading to proton leakage back into the matrix. This situation can also adversely affect the membrane’s structural integrity, crucial for the optimal functioning of protein complexes. Additionally, it has been observed that TPP^+^ derivatives containing an electron-donating group on the aryl ring are not ideal for targeted cargo delivery to mitochondria due to their potent uncoupling of mitochondrial respiration. Conversely, the 4-CF_3_-TPP^+^ derivative effectively delivers cargo to mitochondria in intact cells without impacting mitochondrial respiration or causing toxicity. The notably reduced fungicidal activity of **1C**, in comparison with **1A** and **1B**, suggests that the uncoupling property of the TPP module contributes to the fungicidal activity of type **1** compounds.

The uncoupling of oxidative phosphorylation (OXPHOS) in mitochondria by long-chain fatty acids (LCFAs) has been extensively demonstrated in isolated mitochondria. Due to their proton transport capacity, fatty acids traverse the inner mitochondrial membrane in a protonated state and are subsequently released from the mitochondria as fatty acid anions through anion channels [32]. However, in most cell types, fatty acids act as efficient respiratory substrates. Upon crossing the inner and outer mitochondrial membranes, they undergo β-oxidation in the matrix, transferring electrons to the mitochondrial respiratory chain for energy conversion [33]. Hence, the debate continues regarding whether the in vivo increase in respiration induced by fatty acids is attributed to their intrinsic uncoupling activity within mitochondria or their role as oxidation substrates and stimulation of ATP consumption outside mitochondria. Finichiu et al. observed that the accumulation of TPP-conjugates with weakly acidic or basic groups in mitochondria is governed not only by the membrane potential but also regulated by pH. For example, lipophilic TPP cations linked to undecanoic acid exhibit greater accumulation than simple TPP cations due to interaction with weakly acidic groups and the pH gradient (ΔpH). Conversely, lipophilic TPP cations attached to amines accumulate to a lesser extent than simple cations as ΔpH excludes weakly basic groups [34]. Our study also suggests that the inclusion of TPP preserves the uncoupling ability of fatty acids, although further investigation is required to elucidate its impact on fungicidal activity.

1-Alkanols are well-known for their properties as penetration enhancers and anesthetics [35], with their hydroxyl groups exhibiting an affinity for binding to aliphatic carbonyl groups. The hydrocarbon chains of these alkanols extend into the hydrophobic core of the lipid bilayer, leading to the reduction of the area occupied by each lipid molecule. This reduction increases the thickness of the bilayer, enhances lipid order, decreases bilayer elasticity, and inhibits lateral lipid diffusion within the bilayer plane [36]. The impact of alcohols on lipid membranes is highly influenced by lipid composition, particularly the presence of unsaturated tails and the carbon chain length. Variability in cut-off points observed in protein receptors within neurons may be attributed to different lipid compositions surrounding the receptors [37]. Additionally, Hammond and colleagues have shown that 1-alkyl alcohols can act as uncoupling agents, with the mechanism of respiration inhibition varying based on alkyl chain length [34]. Our research confirms the presence of an uncoupling effect in TPP-C_11_OH. However, further investigation is needed to determine the specific contributions of TPP anion and OH to this activity.

Overall, the fungicidal mechanism of these three type compounds may involve their surface activity, uncoupling effect, and influence on the respiratory chain or other pathways. Deciphering the specific contribution of each activity is a complex task that necessitates further in-depth research.

### 3.5. Mycelium Morphology of P. capsici

Observations of the effects of compounds **1**–**8**, **2**–**12**, and **3**–**11** on the morphology of *P. capsici* mycelia were conducted using an electron microscope, as illustrated in Figure 5. In the control group, the mycelium surface displayed regular and smooth growth with a consistent diameter. Conversely, in the treatment groups of **1**–**8**, **2**–**12**, and **3**–**11**, abnormal mycelial growth was evident. This included increased colony density, overlapping mycelia, surface breakage (highlighted in red), deformation, and nodules (highlighted in blue), as well as folds and atrophy in some mycelia. These findings indicate that the compounds induce varying levels of damage to *P. capsici* filaments, with compound **3**–**11** causing the most substantial damage, followed by compounds **1**–**8** and **2**–**12**. Hence, it can be provisionally inferred that compounds **1**–**8**, **2**–**12**, and **3**–**11** significantly affect the mycelial morphology of *P. capsici*, hindering its normal growth.

### 3.6. Influence of Compounds on Relative Permeability of P. capsici Mycelia

The impact of the compounds on the conductivity of *P. capsici* mycelia were evaluated at a concentration of 17.5 μM, with DMSO (2‰) used as the blank control. The findings are detailed in Appendix A and represented in Figure 6. It was noted that the relative permeability of the mycelium substantially increased within the initial 1–5 h after the agent’s introduction. During this period, the relative permeability of **1**–**8** escalated from 15.65% to 31.85%, **2**–**12** from 7.50% to 17.54%, and **3**–**11** from 12.15% to 26.18%. Subsequently, the seepage velocity gradually declined, reaching a steady state. By the 9 h mark, the final relative permeabilities for compounds **1**–**8**, **2**–**12**, and **3**–**11** were recorded as 37.67%, 20.94%, and 27.83%, respectively. It is noteworthy that these values were all inferior to the control drug fluazinam (45.55%). Furthermore, the 2‰ DMSO led to a rise in relative permeability, consistent with prior reports [38]. Compound **2**–**12** exhibited a relative permeability effect on the mycelium akin to the DMSO control, significantly lower compared to compounds **1**–**8** and **3**–**11**, consistent with the previous fungicidal activity results.

The cell membrane plays a critical role in maintaining cell function, shape, and integrity. Previous research has indicated that the antimicrobial properties of QPS and triphenylphosphonium salts are associated with their ability to disrupt bacterial cell membranes [39]. Similarly, fatty alcohols have been reported to compromise the integrity of bacterial cell membranes. In conjunction with the morphological alterations observed under the electron microscope, it is proposed that compounds **1**–**8**, **2**–**12**, and **3**–**11** impact the cell membrane of the mycelia, leading to compromised integrity. Consequently, the permeability of the cell membrane is disrupted, culminating in an increase in cell membrane permeability. This elevated permeability results in the release of electrolytes from the mycelium, thereby elevating the conductivity of the mycelium. 

Based on the results obtained, it can be deduced that the tested compounds primarily demonstrate fungicidal activity by disrupting membrane integrity. The extent of membrane damage, as reflected by permeability, ranks in the order of **1**–**8** > **3**–**11** > **2**–**12**, aligning positively with their fungicidal capacities. The identified membrane impairment is anticipated to trigger a chain of adverse effects, and the combined influence of these effects ultimately underlies the fungicidal activity demonstrated by the compounds.

### 3.7. Phytotoxicity of Compounds ***1***–***7*** and ***1***–***8***

Due to the potent and broad-spectrum fungicidal effects observed against plant pathogens, compounds **1**–**7** and **1**–**8** underwent additional assessment for their plant safety and potential as agricultural fungicides. However, the data depicted in Figure 7 reveal that at a concentration of 100 μg/mL, both **1**–**7** and **1**–**8** inflicted significant damage on rape leaves and induced minor phytotoxicity on pepper leaves. Therefore, the direct utilization of these compounds as agricultural fungicides is not recommended.

## 4. Conclusions

In summary, our comparative studies with alkyl-TPP^+^ have provided initial insights into the fungicidal activity and mechanism of action of ω-TPP^+^-fatty acids and alcohols. Variances in their surface activity primarily account for the differing levels of fungicidal efficacy observed, while uncoupling activity and other factors may also contribute to a certain extent. The incorporation of carboxyl or hydroxyl groups into alkyl-TPP^+^ can significantly alter its properties, leading to effects such as the inhibition of the respiratory chain complex and the generation of reactive oxygen species (ROS), particularly aggregation behavior, resulting in diminished fungicidal potency. Despite the notable antifungal activity against plant pathogens displayed by compound **1**–**8**, its use as an agricultural fungicide is not recommended due to its phytotoxicity. Alternatively, employing ω-TPP^+^-aliphatic acyl groups and ω-TPP^+^-aliphatic alkoxy groups as targeted delivery systems to mitochondria may cause uncoupling of OXOPHOS and impact the mitochondrial respiratory chain. This complex interplay complicates the role of TPP-conjugates, making it challenging to discern primary from secondary effects. Our findings emphasize that TPP-aliphatic acids or TPP-aliphatic alcohols with a carbon chain length below 10 exhibit minimal fungicidal activity, with their esterified derivatives even less potent. As targeted delivery systems, their individual antifungal properties are overshadowed by the substantial contribution of the TPP moiety to the overall antifungal efficacy of TPP-conjugates. These outcomes lay a robust foundation for designing and developing more effective mitochondria-targeting drugs or pesticides utilizing TPP as a mitochondrial targeting delivery system.

## Figures and Tables

**Figure 1 jof-10-00450-f001:**
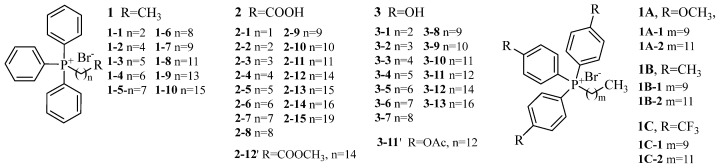
Molecular structures of the synthesized triphenylphosphonium salt.

**Figure 2 jof-10-00450-f002:**
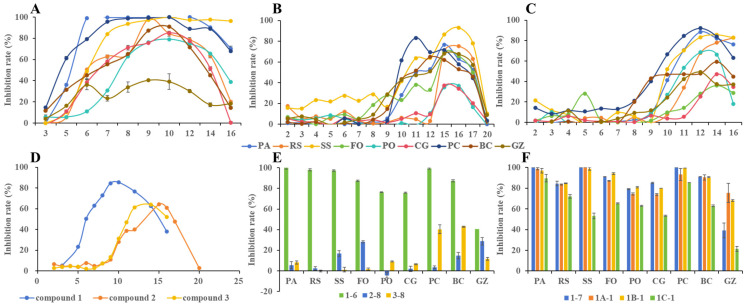
Comparison of fungicidal activities of target compounds. (**A**) fungicidal activity of compounds **1** against different pathogenic fungi; (**B**) fungicidal activity of compounds **2** against different pathogenic fungi; (**C**) fungicidal activity of compounds 3 against different pathogenic fungi. (**D**) Average fungicidal activity of three kinds of compounds against nine pathogenic fungi. (**E**) Fungicidal activities of three kinds of compounds with carbon chain length of 9. (**F**) Impact of substituent effect on benzene ring on fungicidal activity.

**Figure 3 jof-10-00450-f003:**
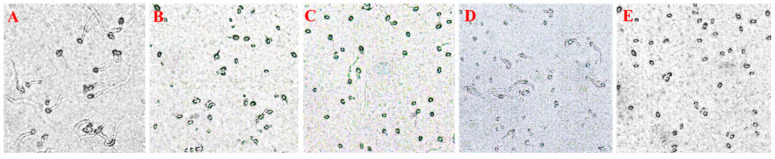
Spore germination inhibition of compound administration against *P. capsici*. (**A**) Blank control; (**B**) 4.375 μM fluazinam; (**C**) 35 μM **1**–**8**; (**D**) 70 μM **2**–**12**; (**E**) 70 μM **3**–**11**.

**Figure 4 jof-10-00450-f004:**
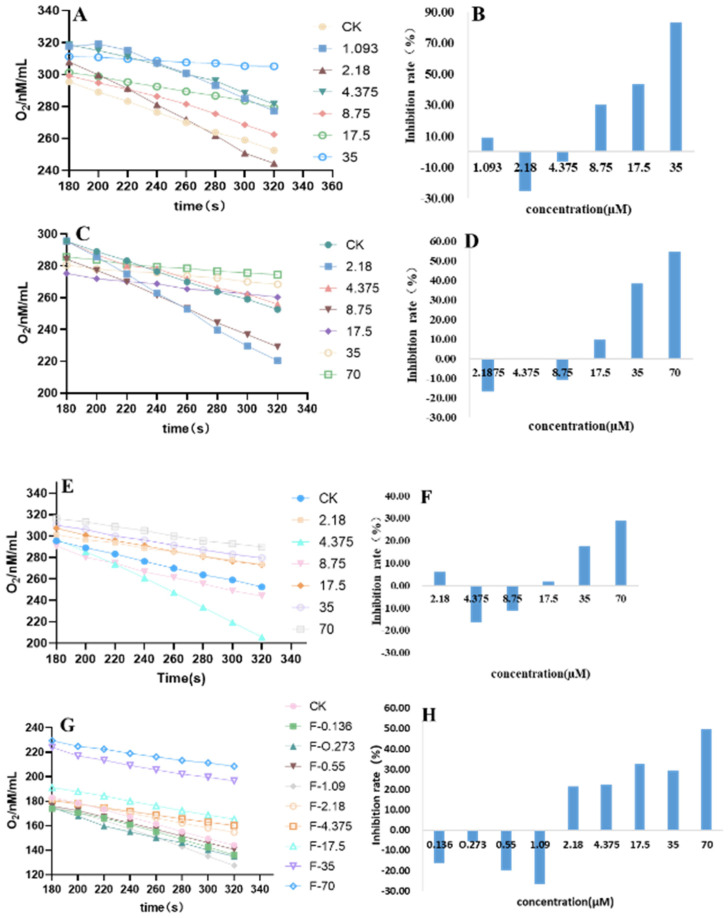
Respiratory curve ((**A**) **1**–**8**; (**C**) **2**–**12**; (**E**) **3**–**11**; (**G**) fluazinam) and respiratory inhibition rate ((**B**) **1**–**8**; (**D**) **2**–**12**; (**F**) **3**–**11**; (**H**) fluazinam) of *P. capsici* mycelia after treatment with agent.

**Figure 5 jof-10-00450-f005:**
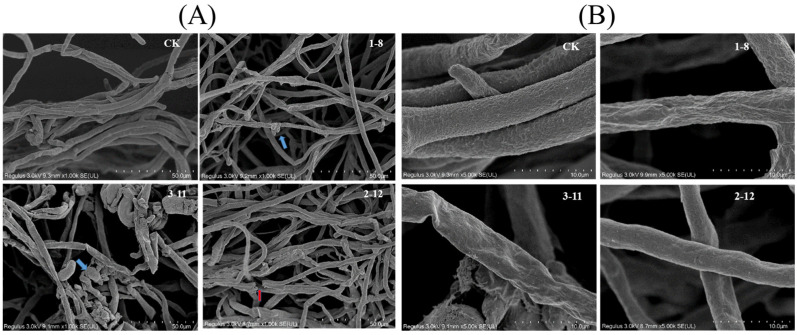
Changes in the mycelial morphology of *P. capsici* after treatment with blank, 35 μM **1**–**8**, **2**–**12**, and **3**–**11** ((**A**) 1000 K; (**B**) 5000 K).

**Figure 6 jof-10-00450-f006:**
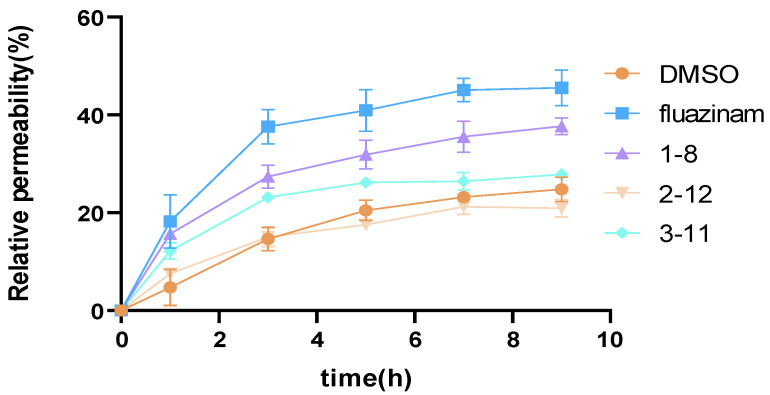
Changes of relative permeability of *P. capsici* mycelia after treatment with agents.

**Figure 7 jof-10-00450-f007:**
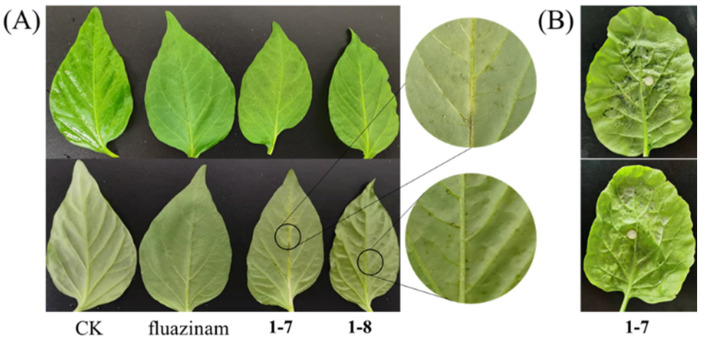
Phytotoxicity of compounds **1**–**7** and **1**–**8**. (**A**) Pepper leaves (sprayed 100 µg/mL solution of **1**–**7** or **1**–**8** and then cultured for 7 d). (**B**) Rape leaves (sprayed 100 µg/mL (**top**) and 200 µg/mL (**bottom**) solution of **1**–**7** and then cultured for 24 h).

**Table 1 jof-10-00450-t001:** In vitro fungicidal activity of compounds **1**, **2** and **3** against 9 pathogenic fungi at 70 μM (%).

Compd.	Inhibition Rate (%)									
	PA	RS	SS	FO	PO	CG	PC	BC	GZ	Mean
**1-1**	0.00 ± 0.00	0.00 ± 0.00	0.00 ± 0.00	11.71 ± 1.27	5.79 ± 1.77	−2.45 ± 2.04	14.67 ± 1.09	11.71 ± 1.27	4.17 ± 1.18	5.07
**1-2**	36.25 ± 1.02	5.42 ± 1.18	10.42 ± 0.59	31.53 ± 0.64	5.79 ± 0.00	11.01 ± 1.36	61.11 ± 1.75	31.53 ± 0.64	16.25 ± 0.00	23.26
**1-3**	99.17 ± 1.18	50.42 ± 0.59	48.96 ± 0.78	45.05 ± 3.19	11.17 ± 1.02	38.43 ± 2.97	79.33 ± 0.94	45.05 ± 3.19	36.04 ± 4.63	50.40
**1-4**	99.58 ± 0.59	62.92 ± 1.18	84.17 ± 0.59	55.41 ± 0.96	30.91 ± 0.59	58.39 ± 2.91	95.56 ± 1.26	55.41 ± 0.96	23.33 ± 2.62	62.85
**1-5**	99.58 ± 0.59	64.58 ± 1.56	93.75 ± 0.00	65.20 ± 0.34	62.99 ± 0.00	71.14 ± 2.04	98.89 ± 0.83	65.20 ± 0.34	33.96 ± 4.85	72.81
**1-6**	100 ± 0.00	98.13 ± 1.35	97.29 ± 1.06	87.39 ± 0.64	76.22 ± 0.78	75.71 ± 0.34	99.33 ± 0.54	87.39 ± 0.64	40.63 ± 0.88	84.68
**1-7**	100 ± 0.00	84.38 ± 2.22	100 ± 0.00	90.99 ± 0.32	79.14 ± 0.51	85.09 ± 0.68	100 ± 0.00	90.99 ± 0.32	39.17 ± 7.24	85.53
**1-8**	100 ± 0.00	78.75 ± 2.04	97.29 ± 0.29	71.62 ± 0.00	74.88 ± 0.29	77.39 ± 0.68	89.11 ± 0.31	71.62 ± 0.00	30.21 ± 1.56	76.76
**1-9**	90.21 ± 0.29	63.75 ± 2.04	97.71 ± 0.59	45.05 ± 1.27	65.68 ± 0.00	51.42 ± 1.80	88.89 ± 0.63	45.05 ± 1.27	17.29 ± 1.93	62.78
**1-10**	70.42 ± 2.12	20.00 ± 1.02	96.46 ± 0.29	14.41 ± 0.64	38.76 ± 0.51	0.43 ± 1.18	68.00 ± 1.09	14.41 ± 0.64	18.33 ± 2.52	37.91
**1A-1**	98.75 ± 1.08	83.49 ± 0.38	100 ± 0.00	87.18 ± 0.36	74.31 ± 1.04	73.96 ± 0.92	93.36 ± 5.85	90.42 ± 2.6	75.35 ± 9.32	86.31
**1A-2**	87.71 ± 2.82	79.14 ± 0.00	97.29 ± 0.95	74.78 ± 1.26	66.80 ± 0.79	70.42 ± 0.68	88.94 ± 1.28	92.92 ± 0.72	19.72 ± 0.00	75.30
**1B-1**	96.88 ± 2.25	84.57 ± 0.38	98.75 ± 1.25	94.12 ± 0.73	80.90 ± 0.68	80.00 ± 0.00	99.26 ± 0.00	90.63 ± 0.63	68.08 ± 0.81	88.13
**1B-2**	81.67 ± 0.72	86.09 ± 1.51	90.00 ± 1.65	77.93 ± 0.63	72.94 ± 0.39	60.00 ± 0.88	87.95 ± 1.54	85.42 ± 1.80	50.23 ± 2.93	76.91
**1C-1**	89.58 ± 3.61	71.97 ± 1.72	53.13 ± 2.72	65.32 ± 0.63	62.94 ± 0.39	53.33 ± 0.51	85.25 ± 0.00	62.92 ± 0.72	21.36 ± 2.47	62.87
**1C-2**	74.79 ± 1.57	52.74 ± 2.30	30.42 ± 1.44	45.57 ± 0.36	44.52 ± 1.97	26.67 ± 0.51	69.27 ± 2.13	40.00 ± 0.00	13.15 ± 2.15	44.13
**2-1**	1.29 ± 0.72	17.50 ± 4.42	15.79 ± 4.96	6.82 ± 2.41	5.09 ± 4.59	5.57 ± 3.00	−1.50 ± 1.19	2.07 ± 0.80	5.62 ± 1.83	6.47
**2-2**	1.50 ± 1.08	6.25 ± 0.00	15.35 ± 1.86	4.08 ± 2.41	4.22 ± 2.24	2.57 ± 1.23	−2.21 ± 0.52	0.64 ± 0.92	7.55 ± 1.88	4.44
**2-3**	−0.37 ± 0.00	7.50 ± 1.88	23.25 ± 1.86	0.18 ± 2.60	5.81 ± 3.28	2.44 ± 0.83	−2.33 ± 1.55	2.07 ± 2.25	4.77 ± 2.66	4.81
**2-4**	−0.17 ± 0.36	5.42 ± 4.07	21.93 ± 4.96	6.94 ± 2.43	8.67 ± 2.59	−0.29 ± 0.92	−3.04 ± 1.79	−4.30 ± 2.39	−0.13 ± 1.78	3.89
**2-5**	0.25 ± 0.62	12.19 ± 3.98	27.63 ± 0.62	9.34 ± 0.00	5.30 ± 5.42	6.11 ± 0.12	5.74 ± 0.64	1.27 ± 0.80	1.88 ± 1.51	7.75
**2-6**	1.50 ± 3.47	3.75 ± 3.80	22.81 ± 2.48	5.68 ± 2.78	1.36 ± 3.93	3.69 ± 1.25	0.22 ± 2.64	−0.85 ± 2.30	4.79 ± 1.03	4.77
**2-7**	1.91 ± 1.44	0.83 ± 3.08	28.95 ± 2.48	18.73 ± 2.10	−1.83 ± 4.41	4.50 ± 2.23	−3.36 ± 2.68	4.46 ± 2.87	9.48 ± 0.91	7.07
**2-8**	5.65 ± 3.43	2.29 ± 2.01	16.84 ± 2.73	28.11 ± 1.05	−4.27 ± 2.52	2.08 ± 2.58	3.57 ± 1.30	14.81 ± 3.18	28.83 ± 3.52	10.88
**2-9**	28.10 ± 2.88	6.25 ± 3.80	41.67 ± 4.34	23.31 ± 2.10	1.34 ± 3.19	5.12 ± 1.40	61.71 ± 1.38	43.47 ± 4.21	43.03 ± 2.41	28.22
**2-10**	50.12 ± 1.87	5.00 ± 4.42	63.74 ± 1.34	37.96 ± 1.05	−0.74 ± 5.44	10.58 ± 0.96	83.36 ± 2.94	51.7 ± 3.59	47.31 ± 5.17	38.78
**2-11**	53.24 ± 3.79	3.13 ± 1.88	64.04 ± 1.24	33.38 ± 1.82	10.69 ± 1.80	9.55 ± 0.95	69.59 ± 0.52	64.97 ± 2.25	50.96 ± 3.69	39.95
**2-12**	76.72 ± 0.36	70.83 ± 1.91	86.55 ± 1.83	69.09 ± 4.86	35.75 ± 2.51	37.29 ± 3.66	71.53 ± 0.89	62.05 ± 2.00	68.32 ± 1.47	64.24
**2-13**	62.93 ± 0.28	75.42 ± 1.57	93.27 ± 4.42	65.89 ± 0.40	37.15 ± 2.10	34.38 ± 1.48	58.05 ± 1.77	53.03 ± 2.25	67.67 ± 1.45	60.87
**2-14**	52.41 ± 2.00	63.13 ± 0.00	78.07 ± 3.51	50.09 ± 2.60	16.53 ± 3.04	20.45 ± 0.45	44.55 ± 1.63	45.86 ± 0.80	57.26 ± 2.05	47.59
**2-15**	0.04 ± 0.72	−1.46 ± 0.95	10.53 ± 3.72	2.70 ± 2.86	−0.56 ± 4.37	3.20 ± 0.39	3.41 ± 1.20	−0.58 ± 4.60	8.98 ± 1.89	2.92
**2-12′**	/	72.52 ± 1.89	83.36 ± 0.80	35.88 ± 0.11	10.02 ± 0.22	15.04 ± 0.70	63.59 ± 0.28	68.90 ± 0.11	25.48 ± 0.15	46.85 ^a^
**3-1**	−4.56 ± 1.02	−4.74 ± 1.66	21.50 ± 0.21	1.76 ± 1.51	−1.70 ± 0.35	1.93 ± 1.17	14.36 ± 0.00	−0.95 ± 1.41	−1.68 ± 1.26	2.88
**3-2**	−0.48 ± 1.01	−1.34 ± 0.92	11.97 ± 0.22	6.38 ± 0.68	10.02 ± 0.71	1.24 ± 1.14	8.45 ± 3.96	−1.81 ± 1.57	0.87 ± 1.93	3.92
**3-3**	0.96 ± 0.43	−5.39 ± 0.21	7.23 ± 0.65	7.64 ± 0.66	−0.81 ± 0.44	6.06 ± 0.50	11.69 ± 3.79	0.82 ± 1.25	11.32 ± 1.11	4.39
**3-4**	−1.68 ± 0.40	4.08 ± 1.84	−1.36 ± 0.35	28.04 ± 0.10	−1.70 ± 4.92	2.20 ± 0.48	10.69 ± 4.32	−1.50 ± 0.82	−1.31 ± 0.73	4.16
**3-5**	−0.89 ± 1.27	4.47 ± 4.43	−1.08 ± 0.76	1.29 ± 2.26	−2.41 ± 0.94	1.00 ± 1.97	13.69 ± 4.36	1.22 ± 1.52	0.27 ± 2.07	1.95
**3-6**	0.86 ± 2.44	−2.21 ± 1.47	9.64 ± 0.86	0.14 ± 0.21	−1.70 ± 1.33	0.86 ± 2.88	13.31 ± 4.53	−0.61 ± 0.58	3.79 ± 1.09	2.68
**3-7**	2.81 ± 0.44	5.03 ± 2.82	5.76 ± 0.51	−0.30 ± 0.58	0.77 ± 2.06	0.50 ± 0.88	21.21 ± 4.78	20.05 ± 1.98	9.26 ± 4.37	7.23
**3-8**	8.20 ± 1.62	−0.15 ± 0.88	1.20 ± 2.13	1.63 ± 1.21	9.09 ± 0.56	6.52 ± 0.47	40.36 ± 4.27	42.93 ± 0.56	11.69 ± 1.38	13.50
**3-9**	41.18 ± 2.10	7.92 ± 3.48	52.10 ± 1.00	9.78 ± 1.00	26.96 ± 1.32	3.87 ± 0.68	66.71 ± 4.17	46.69 ± 2.20	24.44 ± 0.42	31.07
**3-10**	70.82 ± 1.51	34.01 ± 0.84	70.51 ± 0.54	14.23 ± 0.90	53.45 ± 0.84	5.66 ± 0.73	84.52 ± 3.94	47.10 ± 1.80	42.42 ± 0.52	46.97
**3-11**	88.42 ± 1.49	68.00 ± 2.89	83.36 ± 1.80	28.08 ± 0.20	68.55 ± 2.76	25.21 ± 5.27	92.17 ± 4.24	47.76 ± 2.24	49.34 ± 0.52	61.21
**3-12**	82.11 ± 1.38	78.21 ± 0.57	85.68 ± 0.75	36.03 ± 0.60	66.29 ± 0.24	47.04 ± 0.58	83.93 ± 4.06	59.23 ± 1.94	37.32 ± 2.06	63.98
**3-13**	76.28 ± 0.92	82.90 ± 1.95	82.88 ± 1.32	28.91 ± 2.54	18.18 ± 1.29	34.84 ± 2.03	63.29 ± 4.17	44.76 ± 0.65	37.20 ± 3.50	52.14
**3-11′**	/	80.1 ± 2.53	73.94 ± 0.72	42.95 ± 0.23	37.95 ± 0.08	5.65 ± 0.56	86.99 ± 0.28	80.59 ± 0.12	24.53 ± 0.32	54.09 ^a^
**Bs**	3.16 ± 1.30	81.67 ± 0.72	94.44 ± 1.01	29.72 ± 0.79	14.79 ± 5.01	24.52 ± 4.71	27.22 ± 1.00	93.10 ± 1.22	40.32 ± 2.66	45.44
**Km**	28.93 ± 2.65	62.92 ± 0.36	94.44 ± 2.82	50.09 ± 1.05	51.55 ± 0.80	53.60 ± 1.02	22.61 ± 1.21	37.90 ± 2.30	39.11 ± 0.88	49.02
**Flu**	93.14 ± 0.62	100 ± 0.00	95.91 ± 3.96	85.12 ± 3.91	78.57 ± 1.93	99.06 ± 0.28	83.53 ± 0.22	99.63 ± 0.12	99.89 ± 0.08	92.76

PA = *Pythium aphanidermatum*; RS = *Rhizoctonia solani*; SS = *Sclerotinia sclerotiorum*; FO = *Fusarium oxysporium*; PO = *Piricularia oryzae*; CG = *Colletotrichum gloeosporioides* Penz; PC = *Phytophthora capsici*; BC = *Botrytis cinerea*; GZ = *Gibberella zeae*. Bs = boscalid; Km = Kresoxim-methyl; Flu = Fluazinam. ^a^ Average value for eight phytopathogens except *P. aphanidermatum*.

**Table 2 jof-10-00450-t002:** In vitro mycelial growth inhibitory activity (EC_50_, μM) of compounds **1**–**8**, **2**–**12**, and **3**–**11** against *P. capsici*.

Compound	EC_50_	Regression Equation	*R* ^2^	95% Confidence Interval
Fluazinam	0.82	y = 1.63x + 0.13	0.98	0.64–1.05
**1–6**	0.93	y = 0.47x + 0.09	0.96	0.18–2.01
**1–7**	0.90	y = 0.50x − 0.03	0.97	0.19–1.89
**1–8**	0.76	y = 0.94x − 0.16	0.88	0.10–1.87
**1–9**	1.50	y = 0.41x − 0.13	0.97	0.46–2.78
**2–12**	36.73	y = 2.71x − 4.24	0.98	30.16–46.66
**3–11**	6.64	y = 2.06x − 1.70	0.99	5.24–8.26

**Table 3 jof-10-00450-t003:** Inhibitory activity of selected compounds on spore germination of *P. capsici* (IC_50_, µM).

Compound	IC_50_ (μM)	Regression Equation	R^2^	95% Confidence Interval
**1–8**	19.27	Y = −3.03 + 2.39X	0.99	15.60–25.15
**2–12**	70.55	Y = −2.85 + 1.55X	0.99	47.76–136.56
**3–11**	23.42	Y = −5.12 + 3.85X	0.97	18.10–31.19
Fluazinam	1.17	Y = −0.25 + 0.9X	0.97	0.916–1.683

## Data Availability

There no new data were created.

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
