# Peer review of "Phytopathogenic Fungicidal Activity and Mechanism Approach of Three Kinds of Triphenylphosphonium Salts"

_jof, 2024, doi:10.3390/jof10070450_

Round 1
Reviewer 1 Report
The authors describe an interesting study on the biological activity of a diversity of TPP compounds, with a focus on their activity against Phytophthora capsici. The minor corrections are highlighted directly in the pdf file. There are some questions to address in more detail:
1. Add a more precise description in the antifungal activity evaluation in the Materials and Methods section - the comments are in the pdf file.
2. For the evaluation of the phytotoxicity activity, were the compounds dissolved directly in water? You write you sprayed the compounds at 100ug/mL aqueous solution. Or have you dissolved the compounds in DMSO and then diluted them to the required concentration in water? - In that case, what was the final concentration of DMSO? Your control treatment should contain the same concentration of DMSO. If not, were the compounds soluble in water?
3. For a more detailed analysis, why did you not choose compounds 1-7 and 2-10, which show higher activity in P. capsici?
4. Please specify for Table 2 (and corresponding text) if it is still mycelium inhibition test.
5. Figure 4 - unify the y-axis for images A, C, E, G and B, D, F, H for an easy comparison of the activity of the four compounds.
6. The mycelium permeability (Figure 6 and corresponding text) - the compound 2-12 shows activity similar to DMSO (Control). Please comment on that.
7. Phytotoxicity - would it be worth to test lower concentrations of the compounds, as 100ug/ml corresponds ca. to 250uM for the two compounds, if my calculations are right, and the antifungal activity was tested at 70uM concentration.
See the pdf file for detailed comments and corrections

Author Response
- Add a more precise description in the antifungal activity evaluation in the Materials and Methods section - the comments are in the pdf file.
Respond:Thank you for your reminder. We have added a more precise description in the antifungal activity evaluation in the Materials and Methods section.
- For the evaluation of the phytotoxicity activity, were the compounds dissolved directly in water? You write you sprayed the compounds at 100ug/mL aqueous solution. Or have you dissolved the compounds in DMSO and then diluted them to the required concentration in water? - In that case, what was the final concentration of DMSO? Your control treatment should contain the same concentration of DMSO. If not, were the compounds soluble in water?
Respond:Sorry, we didn't make it clear. We diluted the DMSO solution containing the target compound with deionized water to the working concentration, and the content of DMSO was 0.1%. And a water solution with 0.1% DMSO content was used as the blank control. We have added a more precise description in the crop safety assay. in the Materials and Methods section.
- For a more detailed analysis, why did you not choose compounds 1-7 and 2-10, which show higher activity in P. capsici?
Respond:That's a good question. As shown in Table I, compounds 2-12 showed significantly better fungicidal activity than 2-10 against both P. capsici and other pathogenic fungi, and compounds 1-6, 1-7, and 1-8 had comparable fungicidal activity. Thus we further evaluated their IC50 values for inhibition of mycelial growth of P. capsici. Among them, compounds 1-8 were slightly superior to 1-6 and 1-7. The relevant results are not shown in Table 2 and have now been supplemented.
- Please specify for Table 2 (and corresponding text) if it is still mycelium inhibition test.
Respond:Yes, it is still mycelium inhibition test. We have added detailed explanations in Table 2.
- Figure 4 - unify the y-axis for images A, C, E, G and B, D, F, H for an easy comparison of the activity of the four compounds.
Respond:Thank you for your suggestion. We tried to standardize the vertical coordinates of these images, but due to significant differences, the standardized images were not clear and beautiful (as shown below). All things considered,we did not make any changes.
- The mycelium permeability (Figure 6 and corresponding text) - the compound 2-12 shows activity similar to DMSO (Control). Please comment on that.
Respond:That's a good question. 2‰ DMSO caused an increase in relative permeability, which is consistent with other article reports.
For example, as in the ref. https://doi.org/10.1021/acs.jafc.2c03163
- Phytotoxicity - would it be worth to test ower concentrations of the compounds, as 100ug/ml corresponds ca. to 250 µM for the two compounds, if my calculations are right, and the antifungal activity was tested at 70uM concentration.
Respond:That's a good question. 100 µg/ml corresponds to 200 µM for compounds 1-7 and 1-8. And these compounds have an in vitro IC50 of approximately 1µM against P. capsici. However, when used in the field, the effective concentrations of the compounds were much higher than the in vitro experimental concentrations. For example, The IC50 of fludioxonil against P. capsici was 0.85 µM by the mycelial growth inhibition assay. And the recommended dosage of 50% fludioxonil suspension for the treatment of P. capsici in the field is 25-35 mL/667m2, which is equivalent to a concentration of 100-600 µg/mL.

Reviewer 2 Report
The present manuscript is generally well-written and illustrated. I think this work is suitable and within the scope of the Journal of Fungi and it will be useful for the broad readership of the Journal of Fungi. The following minor details for improvement or consideration can be pointed out:
The present manuscript is generally well-written and illustrated. I think this work is suitable and within the scope of the Journal of Fungi and it will be useful for the broad readership of the Journal of Fungi. The following minor details for improvement or consideration can be pointed out:
1) References should be given in accordance with the Instructions for the authors.
2) The structures of the synthesized compounds should be clear from Fig. 1. What is 1-1, 1-2, 1-3, etc.
For example, as in the Scheme 2, ref. https://doi.org/10.1039/d0ra06263d.
3) Line 165 “method yielded three compounds, namely 1A, 1B, and 1C.” Really, six compounds were synthesized.
4) Lines 162, 209: “condensed” should be changed to “evaporated”.
5) Line 208 “ω-bromoalkanic” should be changed to “ω-bromoalkanoic”.
6) Line 419, 491 “triphenylphosphane" Triphenylphosphine is a correct systematic name.
7) Lines 678-679 “The results are presented in Table S1” Table S1 is not available in the manuscript or supplementary data. (Lines 752-753 “The Supporting Information is available free of charge at https:// 1H NMR, 13C NMR, and HRMS spectrum data of target compounds (PDF)”)
8) Lines 681-682: “Specifically, within this time frame, the relative permeability of 1-8 increased from 15.65% to 40.92%” According to Fig.6, within the first 1-5 hours the relative permeability of 1-8 increased to ca. 30%.
9) Lines 725-726: References in the Conclusions chapter are inappropriate.
10) I believe that the chemistry part of Synthetic Procedures is too extensive for the Journal of Fungi and it would be desirable to move the data on the compounds obtained to the Supporting Information section.
11) Errors in the description of NMR and HRMS data:
Line 114 “calcd. for C24H28P [M-Br]+ 333.1767, found333.1761.” Incorrect data.
Lines 120, 126, 132, 138, 144, 151, 158, 233, 239, 251: Incorrect molecular composition for calculated HRMS.
Line 292: “ calcd. for C34H48OP [M-Br]+ 503.3437, found 503.3440” Incorrect data.
Line 143: “22.68 (d, J = 4.8 Hz), 22.68,” This is true?
Lines 196, 203: “127.46 (dd, J = 13.5, 3.5 Hz)” This is true? This signal should appear as dq.
Line 219 “133.75b” should be corrected.
Line 225: “22.55 (dd, J = 18.6, 3.6 Hz)” This is true? This signal should appear as d.
Line 269 “118.01 (dd, J = 85.8, 6.2 Hz),” This is true? This signal should appear as d.
Line 320: “1.16 (d, J = 44.2 Hz, 18H)” Incorrect data.
Line 345: signal 171.78 is incorrect for this compound. Also, signal of CH2O group ca. 60 ppm is omitted.
Line 376, 382: data for one more CH2 group ca. 3.6 ppm are omitted.
Line 424: “2.04 (, 3H)” should be corrected.
Author Response
1) References should be given in accordance with the Instructions for the authors.
Respond:Thank you for your reminder. We have downloaded the MDPI.ens references style file from the EndNote website according to the journal prompts and modified the format of all references using EndNote software.
2) The structures of the synthesized compounds should be clear from Fig. 1. What is 1-1, 1-2, 1-3, etc.
For example, as in the Scheme 2, ref. https://doi.org/10.1039/d0ra06263d.
Respond:Thank you for your suggestion. We have made corrections in the article.
3) Line 165 “method yielded three compounds, namely 1A, 1B, and 1C.” Really, six compounds were synthesized.
Respond:Thank you for your reminder. It should be “method yielded three types of compounds, namely 1A, 1B, and 1C.” We have made changes in the article.
4) Lines 162, 209: “condensed” should be changed to “evaporated”.
Respond:Thank you. We have made changes in the article.
5) Line 208 “ω-bromoalkanic” should be changed to “ω-bromoalkanoic”.
Respond:Thank you very much! We have revised it
6) Line 419, 491 “triphenylphosphane" Triphenylphosphine is a correct systematic name.
Respond:Thank you again. We have made changes in the article.
7) Lines 678-679 “The results are presented in Table S1” Table S1 is not available in the manuscript or supplementary data. (Lines 752-753 “The Supporting Information is available free of charge at https:// 1H NMR, 13C NMR, and HRMS spectrum data of target compounds (PDF)”)
Respond:Thanks! Table S1 is located at the end of the supplementary data.
8) Lines 681-682: “Specifically, within this time frame, the relative permeability of 1-8 increased from 15.65% to 40.92%” According to Fig.6, within the first 1-5 hours the relative permeability of 1-8 increased to ca. 30%.
Respond:Thank you for your reminder. The data on lines 681-682: “Specifically, within this time frame, the relative permeability of 1-8 increased from 15.65% to 40.92%” is incorrect. We have verified and corrected this.
9) Lines 725-726: References in the Conclusions chapter are inappropriate.
Respond:Thank you for your reminder. We have made verifications and corrections.
10) I believe that the chemistry part of Synthetic Procedures is too extensive for the Journal of Fungi and it would be desirable to move the data on the compounds obtained to the Supporting Information section.
Respond:Thank you for your suggestion. We have moved the NMR data of compounds to the Supporting Information section.
11) Errors in the description of NMR and HRMS data:
Line 114 “calcd. for C24H28P [M-Br]+ 333.1767, found333.1761.” Incorrect data.
Respond:Thank you for your reminder. We have made verifications and corrections.
Lines 120, 126, 132, 138, 144, 151, 158, 233, 239, 251: Incorrect molecular composition for calculated HRMS.
Respond:Thank you for your reminder. We have made verifications and corrections.
Line 292: “ calcd. for C34H48OP [M-Br]+ 503.3437, found 503.3440” Incorrect data.
Respond:Thank you for your reminder. We have made verifications and corrections.
Line 143: “22.68 (d, J = 4.8 Hz), 22.68,” This is true?
Respond:Thank you for your reminder. We have made verifications and corrections.
Lines 196, 203: “127.46 (dd, J = 13.5, 3.5 Hz)” This is true? This signal should appear as dq.
Respond:Thank you for your reminder. This should indeed be the dq peak, but due to overlapping, we have changed it to the m peak in the article.
Line 219 “133.75b” should be corrected.
Respond:Thank you for your reminder. We have made verifications and corrections.
Line 225: “22.55 (dd, J = 18.6, 3.6 Hz)” This is true? This signal should appear as d.
Respond:Thank you for your reminder. It should be “13C NMR (126 MHz, DMSO-d6) δ 173.29 , 134.95 (d, J = 3.3 Hz), 133.58 (d, J = 10.1 Hz), 130.27 (d, J = 12.4 Hz), 118.37 (d, J = 86.2 Hz), 33.64 (d, J = 18.1 Hz), 19.84 (d, J = 51.0 Hz), 17.74 (d, J = 3.6 Hz).” We have made verifications and corrections.
Line 269 “118.01 (dd, J = 85.8, 6.2 Hz),” This is true? This signal should appear as d.
Respond:Thank you for your reminder. We have made verifications and corrections.
Line 320: “1.16 (d, J = 44.2 Hz, 18H)” Incorrect data.
Respond:Thank you for your reminder. We have made verifications and corrections.
Line 345: signal 171.78 is incorrect for this compound. Also, signal of CH2O group ca. 60 ppm is omitted.
Respond:Thank you for your reminder. We have made verifications and corrections.
Line 376, 382: data for one more CH2 group ca. 3.6 ppm are omitted.
Respond:Thank you for your reminder. This peak was overlooked by us due to its overlap with the water peak. We have made verifications and corrections.
Line 424: “2.04 (, 3H)” should be corrected.
Respond:Thank you for your reminder. We have made verifications and corrections.
